# Dynamics of Strength Gain in Sandy Soil Stabilised with Mixed Binders Evaluated by Elastic P-Waves during Compressive Loading

**DOI:** 10.3390/ma15217798

**Published:** 2022-11-04

**Authors:** Per Lindh, Polina Lemenkova

**Affiliations:** 1Swedish Transport Administration, Neptunigatan 52, P.O. Box 366, SE-201-23 Malmö, Sweden; 2Division of Building Materials, Department of Building and Environmental Technology, Lunds Tekniska Högskola (LTH), Faculty of Engineering, Lund University, P.O. Box 118, SE-221-00 Lund, Sweden; 3Laboratory of Image Synthesis and Analysis, Building L, École Polytechnique de Bruxelles, Brussels Faculty of Engineering, Campus de Solbosch, Université Libre de Bruxelles, ULB–LISA CP165/57, Avenue Franklin D. Roosevelt 50, B-1050 Brussels, Belgium

**Keywords:** civil engineering, stabilisation, compressive strength, soil, cement, slag, fly ash, lime, seismic waves, 81.40.Cd, 81.40.Ef, 62.20.Qp, 83.50.Xa, 45.70.Mg, 92.40.Lg, 81.40.Lm, 62.20.M–, 76Axx, 74Exx, 74Fxx, Q00, Q01, Q24, Q55, Q56

## Abstract

This paper addresses the problem of stabilisation of poor subgrade soil for improving its engineering properties and stiffness. The study aim is to evaluate the effects from single and mixed binders on the gain of strength in sandy soil over the period of curing. We propose an effective non-destructive approach of using P-waves for identifying soil strength upon stabilisation. The growth of strength and stiffness is strongly dependent on time of curing and type of the stabilising agents which can include both single binders and their blended mixtures. The diverse effects from mixed binders on the properties of soil were evaluated, compared and analysed. We performed the experimental trials of five different binders for stabilisation of sandy soil using cement, lime, Ground Granulated Blast Furnace Slag (GGBFS), energy fly ash and bio fly ash. The methodology included soil stabilisation by binders during a total period of 90 days, strength test for the Unconfined Compressive Strength (UCS) and seismic tests on the stabilised samples. The dynamics of soil behaviour stabilised by different binders for days 7, 14, 28 and 90 was statistically analysed and compared. The optimisation of binder blending has been performed using mixture simplex lattice design with three binders in each case as independent variables. Using P-waves naturally exploited strength characteristics of soil samples and allowed us to compare the effects from the individual and blended binders over the complete period of curing with dominating mixes. The results indicate that strength growth in stabilised soil samples is nonlinear in both time and content of binders with dominating effects from slag which contributed the most to the compressive strength development, followed by cement.

## 1. Introduction

Soil stabilisation which can be defined as an improvement and refinement of soil parameters through modification of its engineering properties is currently being used in many of the state-of-the-art high-level civil engineering tasks such as construction works, maintenance of built environment, including structural components of buildings, roads, bridges, airports, etc. The goal of laboratory experiments on soil stabilisation, given a context of road and highway construction, is to determine the compaction characteristics and the strength of the stabilised material. Stabilisation of soil with traditional binders is used in a wide variety of engineering applications with the major aim at improving soil properties prior to construction works in road engineering [1,2,3,4,5,6,7]. Cement and lime are the oldest, fundamental and most well-known traditional binders, continuously used in civil engineering [8,9,10,11]. Therefore, their applications for soil stabilisation have been researched and reported for decades as a straightforward, versatile and traditional method of soil improvement [12,13,14,15,16].

Many types of binders have been tested in industrial works and proposed in the literature that can be broadly classified into two groups—traditional binders and novel mixes of blended binders. Although traditional binders, such as cement or lime, remain the most commonly used additives in soil stabilisation, the selection of binders becomes more complicated owing to the recent issues. These include negative environmental impacts from cement, the increased amount of the industrial by-products which should be utilised and the fabrication of novel stabilising agents which have improved geotechnical properties in terms of engineering performance and better suited to the ecological challenges. For instance, such cases arise when geotechnical works are planned on the expansive or clayey soils that have specific properties [17]. Moreover, compared to the traditional binders, blended binders have been shown to produce more consistent effects on soil stabilisation by adaptively balancing between the effective refinement of soil parameters and economic costs of works. Blended binders, combined from several materials of various additives can respond to such needs and requirements, since they are better suited to different types of soil and regional environmental conditions, thus considering real project conditions.

In this regard, blended binders can be considered as an advanced method of soil stabilisation which has been shown to perform better than traditional binders in the state-of-the-art techniques of soil stabilisation [18,19]. Although it derives soil stabilisation results by improving its parameters, which leads to an increased strength of foundation and durability of structures, the use of traditional binders is restricted in the environmental and technical performance. The experimental use of the alternative and blended binders allows for application of more effective approaches, both in economical and engineering aspects [20], which is valuable, due to the high cost of geotechnical works on road construction.

Various attempts have been made to understand how the addition of of blended binders can enhance soil properties and increase its strength, and in which proportions binders should be taken [21,22,23,24,25,26]. Indeed, mixed binders have the potential to improve soil stabilisation through more appropriate workflow of the experiments aimed at reducing soil permeability, compressibility, deformation and settlement. For instance, the use of blended binders supports the effectiveness of clayey soil hardening over time of curing [27,28,29].

Specifically for Sweden, the performance of single and mixed binders was investigated using deep mixed method [30,31]. The most well-known application of blended binders in Sweden is the depth stabilisation technique, which follows the long-term tradition of soil treatment in the country [32]. Others include using blended binders for shallow soil stabilisation. Stabilisation has also gained increased use in infrastructure projects for treatment of the contaminated soil [33,34]. Further, blended binders perform well compared to single binders, because such mixtures are better suitable on soils which with varied grain size [35,36]. Since the response of such soils varies to the different types of binders, blended mixes can be applied for testing soils with diverse grain content, ranging from a fine-grained clay to a coarse gravel. Moreover, the use of mixed binders in Sweden can help make cemented soil structure estimates consistent with the in situ conditions of real projects, such as road engineering in cold environment with cases of ground freezing.

As the areas of applications increase along with the use of complex mixtures of binders, so do the requirements for the optimisation of soil. In the post-treatment of the contaminated soil, it is common for chemically active additives to be used to increase the effect of stabilisation with regard to the reduction of the environmental risk. It is also common for cement or slag to be replaced by other binders, such as fly ash or other residual products, with aim to reduce costs of engineering works [37,38,39]. Moreover, in response to the increased societal demands for environmentally sustainably solutions in engineering works, e.g., reduced CO${}_{2}$ emissions, the demands on the environmental impact from binders during the process of soil stabilisation are also increasing. Overall, these requirements and new challenges in road construction industry lead to the need to gain access to novel methods and materials that enable optimisation of soil treatment that would take into account both technical performance, economics and environmental aspects which are to be considered in the final project.

The Uniaxial Compressive Strength (alternatively: Unconfined Compressive Strength) (UCS) test is a straightforward and relatively easy method to evaluate the enhancement of strength and stiffness in stabilised soil over time. Because it well reflects both compaction properties and gained strength, it is one of the most frequently used and commonly acceptable methods to evaluate the engineering properties of the stabilised soil, as continuously reported in earlier papers [40,41,42,43]. The examples of using the UCS tests are diverse and include, for instance, the improved methods testing bearing capacity and settlement of cohesive soil in the foundation of buildings [44]. Some more advanced methods include predictive modelling by programming techniques to estimate the UCS [45], using artificial intelligence methods [46,47,48] and machine learning applications [47,49] to UCS estimation. Almost all previous UCS-based soil testing techniques have been restricted to existing workflow using traditional binders. However, the major disadvantage of the UCS method consists in its destructive nature, since it crushes the tested sample during the experiment.

Therefore, the proposed framework uses seismic measurements which alleviates the above difficulties and disadvantages of of the UCS by measuring soil strength using elastic P-waves through non-destructive measurements. The seismic testing presents an alternative method of evaluation of stiffness and strength parameters of soil stabilised by various percentages of binders which can be performed using measured velocities of elastic pressure waves (P-waves). The core principle of the evaluation of velocities of P-waves consists in the physical theory of waves and elastic properties of soils as a porous media. Thus, measuring P-wave speed enables to estimate the level of stiffness and strength in a soil sample, as used in various geotechnical application [50,51,52,53,54]. The technical approach of this method consists in the propagation of elastic waves that penetrate the specimens of soil materials tested at different curing periods. A quantitative assessment of the P-wave velocity gives the value of the soil strength and stiffness due to the correlation between these parameters, which is reported in relevant case studies [55,56,57,58]. Seismic methods applied to measuring the properties of soil demonstrated superiority over common methods due to its non-destructive nature, which enables to measure samples as many times as needed, but requires complex tools and advanced methods of data processing [59,60].

This paper deals with experimental trials of five different binders for stabilisation of clayey soil: cement, lime, GGBFS and two types of fly ash (energy and bio fly ash). The combinations of blended binders were determined using special techniques of simplex design, aimed at optimising the proportions of the stabilising agents. The methodology comprises soil treatment by blended binders during the overall period of 90 days, and geotechnical tests according to the standards of the Swedish Geotechnical Institute (SGI), Linköping: strength test for the UCS and seismic measurements (P-wave speed) of the stabilised specimens. The dynamics of soil behaviour stabilised by different binders was statistically analysed and compared on the 7th, 14th, 28th and 90th days, respectively. The development in P-wave velocity was compared both for single and blended binders over the complete period of curing, i.e., 90 days. The results are summarised in tables and visualised on plots in relevant chapters below. Our main contribution can be summarised as follows: (1) the approach of P-wave tests that performs non-destructive measurements of soil strength in selected samples during the period of curing; (2) stabilisation using both single and blended binders; (3) using statistical simplex design approach as an associated combinatorial structure and factor experiment for optimisation of binder quantity and water ratio; and (4) the use of statistical methods for data analysis based on measurements of P-wave velocities as a function of curing time for individual and blended binary binders.

## 2. Materials and Methods

### 2.1. Materials

#### 2.1.1. Specimens

Soil used in this study was excavated from the selected places in southern Sweden. After collecting and refining, the specimens were processed in the laboratory of SGI. Afterwards, the soil grain size was examined according to the grain size distribution. Based on the examined soil, its structure is dominated by the two major types: sand and gravel, according to the grain size (Figure 1).

The soil used in this study has a coarse structure and includes fine to medium silty sand and gravel inclusions obtained in the study area of southern Sweden. According to the Unified Soil Classification System (USCS) the sandy soil samples can be classified as follows: clayey sands, sand-clay mixtures (SC) up to 10% of samples, silty sands, sand–silt mixtures (SM) and poorly graded sands (SP) (from 10% to 18% of samples), and well-graded sands, gravelly sands (SW) (18–25% of sample) and belongs to group sands with half or more of coarse fraction, Figure 1. The first group of gravel specimens (25–50%) is presented by clayey gravels and gravel-sand-clays (GC) and silty gravels, gravel-sand-silt mixtures (GM), followed by medium gravel (over 50%), characterised by poorly-graded gravels, gravel-sand mixtures (GP), see Figure 1.

The grain size of sand is ranging between the 0.06 to 20 mm (sand) and for 2.0 to 31.5 mm for the gravel. The ballast material (support layer) was weighed in three batches of 1700 g each. Water was measured in measuring glasses, so that the corresponding 102 g per batch weighed had a water ratio of 6% before the addition of binder and 5.8% when the binder was included. To ensure the significance level of the results, the recipes of binder-soil mixtures were based on the previous experience of similar tests in SGI in combination with the statistical experimental tests (simplex). The statistical trial design minimises the number of trials that need to be performed with the desired level of significance. Thus, for instance, for the five different binders, we fabricated approximately 100 soil specimens.

The specimens were stabilised by binders and stored for 90 days of curing period to achieve final strength, even for the slow-hardening components. During this time, the velocity of the P-waves and in selected cases also the speed of the shear wave have been measured in the material at different times. The long curing period of up to 90 days, was selected, because it is the one of the key factors that affect soil stabilisation. After the 90th day of storage, selected samples were reduced in size twice, after which the plastic tube was removed. Long period of curing ensures the improvement of strength which tends to increase with time due to the bonded particles in a soil-binder mixture. Moreover, the development of soil-lime pozzolanic reaction increases over time.

#### 2.1.2. Binders

The stabilisation of soil by binders was aimed to improve its geotechnical characteristics prior to construction works. Specifically these include the following goals: to reduce and prevent settlement of soil as a road basement; to increase soil strength; to improve bearing capacity of soil; to lower soil permeability; to prevent shrinkage when undergoing changes in moisture (water) content; to control volume changes in soil caused by the thaw-freeze effects, which is a subject to environmental and climate effects, as often the case for Sweden. To this end, we used five different binders for soil stabilisation: two traditional binders (ordinary Portland cement (OPC) and lime) and three alternative binders (slag GGBFS, bio fly ash and energy fly ash), originated from the SCA Lilla Edet and the from coal combustion, certified by International Organization for Standardization (ISO).

The GGBFS slag is of type Merit 5000, which is an additive material for cement and concrete made from dried Hyttsand. Merit 5000 has been used in combination with OPC due to similar effects on strength development in soil [61]. The bio fly ash originated as a by-product from the CHP power plants that use only biomass as fuel where pure bio fly ash is separated from the gas in electrofilters before the gas is directed to the chimney. The bio fly ash was used as an alternative binder and a component to binder blends in this study. The first fly ash origin is from Svenska Cellulosa Aktiebolaget (SCA) Lilla Edet and the second is from coal combustion (ISO certified). The bearing layer gravel (0–18) is used as a ballast material. The choice of these binders was approved for the three reasons.
First, these binders were selected based on the grain size of the specimens, to be suitable to the material used in a real project, where a maximum particle size reached 20 mm (from fine sand to course gravel).Second, it was necessary to use a material with small variation in physiochemical properties, which would minimise the fluctuations in final stabilisation results.Third, the additives should be inert and not contribute to the strength gain in combination with any other binder. In the first step, sample specimens were fabricated and stabilised using different binder recipes.

Detailed schedule of the binder recipes for each soil specimen is presented in Table A1.

### 2.2. Workflow

The workflow included the evaluation of the effects from single and mixed binders on the gain of strength in soil specimens over time of curing. The experimental trials included treatment of soil by five different binders used for stabilisation of clayey soil: OPC, lime, GGBFS, energy fly ash and bio fly ash. The optimisation included simplex design combinatorics method for correct selection of binder combinations and amounts. The experimental planning aimed at reducing the financial risks of the project, since large amounts of soil had to be stabilised in a real project (several hundreds of tons of soil).

The optimisation of binder blending was performed in the laboratory of SGI using modelling by mixture simplex lattice design with three binders in each case as independent variables. The performed methodology include soil treatment by 5 types of binders during 90 days, and geotechnical tests according to the SGI standards: strength test for the Unconfined Compressive Strength (UCS) by the SIS standard SS-EN 13286-41 and seismic tests on the stabilised samples using ICP Accelerometer for measuring P-wave velocity. The dynamics of soil behaviour stabilised by different binders for days 7, 14, 28 and 90 was statistically analysed and compared. The curves of strength gain in soil specimens have been compared using various ratios of binders (double and triple combinations of stabilising agents). Furthermore, the variation in water ratio in specimens before and after the complete curing period were statistically assessed.

#### 2.2.1. Uniaxial Compressive Strength (UCS)

The UCS tests were carried out on stabilised soil using the 10-ton pressure press, Figure 2a. The tests were run under the deformation control following the existing standard methodology SS-EN 13286-41 [62,63]. The UCS was performed to achieve the maximum axial compressive stress that a specimen can bear under zero confining stress. In such a way, the specimens were tested using the available equipment at SGI (Figure 2a,b for schematic and real view, respectively). A workflow methodology is based on the the Swedish Institute for Standards (SIS) [62,63]. The breaking load reached 60 s, which was achieved at a deformation rate of 3–4 mm/s, Figure 2b. No visible connections between the bearing boundaries and fracture surfaces were noted on any of the tested specimens. The specimens exhibited the hourglass-shaped fracture surfaces that were completely independent of the layer boundaries.

#### 2.2.2. P-Wave Measurements

During the curing period of 90 days, measurements of the resonant frequency of the elastic waves were performed on soil specimens on days 7th, 14th, 28th and 90th using free-free resonate column approach. The velocity of the pressure waves (P-waves) was measured in specimens on control days during curing period, to evaluate the dynamics of strength gain. The P-wave velocities of the stabilised soil were received using the Integrated Circuit Piezoelectric (ICP) Accelerometer (Model Nr. 352B10), a lightweight ceramic shear response device designed by the PicoCoulomB (PCB) Piezotronics https://www.pcb.com. The workflow included the following steps. An impulse force was excited in a tested specimen, and the velocity of pressure wave was recorded by the device. The first arrival time of wave was estimated to compute the travel time of the compression wave. The velocity of the pressure waves was computed using travel time of the pulse propagated through the soil specimens. The data were then transferred to the computer for statistical processing.

#### 2.2.3. Determination of Water Content

The determination of water content has been performed using the existing methodology [64]. The variation in water ratio between the specimens before and after the curing time of 90 days is shown in Figure 3 which illustrates this difference. Here the blue colour shows water content before curing, while the red colour shows water content after the 90 days of curing. Since samples were paraffined, no water remained in soil samples, but the difference in water ratio can be explained by chemical reactions during curing.

Target water ratio for the stabilised material was 5.8% when binder was included. The histogram in Figure 3 illustrates the mean pre-storage of 5.89% with a standard deviation of 0.307; after curing the mean was 5.37 and the standard deviation −0.359, which indicates that the target value was achieved. The variation in water content was not greater than expected, which depends, i.a., on the composition of the material used for water ratio determination. In samples with a low water ratio, there have been coarser materials and vice versa for those with a high water ratio. This is because coarser grains cannot bind as much water as several smaller grains together can. The measurement was performed as a quality control to discard any statistical outliers. For these reasons, two samples were discarded due to the incorrect values of water quotas. The samples reported in Figure 3 only include those accepted after statistical evaluation.

#### 2.2.4. Freeze-Thaw Tests

The freeze-thaw tests were performed according to the Swedish technical standard SS-EN 137244 [65]. In this method, sample specimens were frozen for a total period of 56 days. The amount of material was measured on days 7th, 14th, 28th, 42th and 56th. The temperature varied and demonstrated the dependance on curing time. No extra water was added during these cycles to ensure the objectivity of the experiment. The samples used for the freeze-thaw tests were selected from the original soil mass in connection with the UCS tests. Therefore, the specimens that did not hold together could not be tested, for instance, the specimens stabilised by energy fly ash or bio flay ash in large amounts which did not stick together. These tests showed that all tested samples had a high frost resistance.

The processes of freezing and thawing, as common environmental and climate induced phenomena in Sweden, often lead to the increase in volume of soil and reduction of strength. Hence, it is important to consider durability of soil as the final goal with regard to the improved soil properties prior to the road construction. At the same time, freeze-thaw test results may fluctuate and differ, which depends on the actual field conditions. This is owing to the placement of the stabilised soil in the in situ conditions below the road pavement. As a result, changes in moisture condition in soil sample may vary from saturated to dry. Likewise, stabilised soil samples may not be frozen when saturated due to the local climate specifics in southern Sweden.

### 2.3. Concepts

The conceptual optimisation of the workflow using experimental design is an important part of geotechnical works, as it enables to minimise the number of tests. In view of the large amounts of material that should be tested (hundreds of tons), the optimisation of work is necessary to plan the workload in a more economically effective way. To assess the impacts from the individual and mixed binders, a simplex mixture design was chosen following the existing methodology [66], Figure 4.

The role of the trial planning consists of a systematic methodology aimed at finding the optimal and minimal necessary content of the tested binder products which should be added into the soil mixture. The traditional method to optimising binder mixtures uses the approach which consists in one factor tested at a time. This strategy enables only one parameter to be changed, while keeping all the other parameters as constant values. However, by this approach, there is a risk of missing both positive and negative interactions between the constituent components of the soil-binder mixture. Neither does it allow a simultaneous optimisation which could be based on the analysis of both technical and environmental requirements of the stabilised soil.

The decision regarding optimal binder mixture is based on a series of different random trials where various factors can be varied in an unsystematic way. The mentioned above problems can be eliminated if the statistical trial planning is used to support the optimisation of binder mixtures. Thus, trial design ensures that there is a systematic evaluation of binder mixtures, which is achieved through simultaneous considering of the influence of various factors on the final results [67]. Specifically for soil testing, one can measure the effects of various binder combination on the evolution of soil strength over the period of curing. By working systematically, the possibility of making a good optimisation of blended mixtures of binders is increased with regard to both technical performance, economic gain and environmental impact.

Laboratory tests on binder proportions and combinations often entail higher costs due to many trials in the initial stage. Using this method, high total cost of the geotechnical works on soil stabilisation intended for road construction can be significantly reduced. Since both the choice of binder components and binder quantities largely affect project economy, the price of works can be reduced in relation to the real needs. Therefore, to optimise binder mixtures we used modelling trial design techniques. The approach can be divided into a mixing part and a process control part. Mixture optimisation included the mutual proportions of different binders in the final mixture, that is, the ratios of blended binders [68]. The process control optimisation stage focuses on how much binder should be used based on the properties of the base soil material, e.g., how much water should be added to balance the ratio and, if necessary, what is the level of contamination as environmental process parameters.

#### 2.3.1. Optimization of Binder Mixture

For the two binders, the optimisation process is a relatively simple and straightforward process. In this case, all the possible binder combinations can be placed on a straight line that can be described according to Equation (Equation 1), Figure 4a. As long as x<1 or y<1, the mixture consists of two parts. If x=1 or y=1, the mixture consists of a single binder component.
(1)y=1−x

Possible binder combinations with two different binders are represented by a straight line where the endpoints signify one binder, 1.0 = 100%, i.e., a one-component mixture. Respectively, adding the 2nd binder automatically reduces the amount of the 1st binder, so that together they always constitute a total amount of 100%, see Figure 4a. However, for mixtures consisting of three or more binders, the situation becomes more complicated, as the optimal final mixture is hidden among the infinite number of possible mixture combinations. The experimental planning handles this case using the simplex method, initially developed in statistical works [66,67]. The general aim of simplex method is to evaluate different types of mixtures with three or more constituent components, which is a very common optimisation scenario in real case situations in industry. Figure 4b illustrates a graph representing all possible mixing ratios for the three components A, B and C, based on the simplex method. The corners represent 100% of a single binder. The border lines represent mixtures between the two different binders, according to Equations (Equation 2)–(Equation 4).
(2)y=1−x
(3)z=1−y
(4)x=1−z

The plane bounded by the edge lines consists in the infinitely many combinations of all three binders which can be experimentally mixed and tested. All the possible binder combinations with the three different binders (A, B and C) are represented by a plane bounded by the three straight lines (Figure 4b). Here each vertex in the plane represents a single binder, i.e., 1.0 = 100% of a single component [66]. Depending on the scale and amount of works chosen for simplex method, both linear and interaction effects can be identified. These include, for instance, the interaction effect which occurs when two binder components together produce a significantly higher or lower effect on soil strength than their mutual proportions indicated earlier. The level of details, however, places different demands on the total number of tested trials. Therefore, the amount of the experimental trails should always be reasonably high. The optimal combination of binders is required for the effective soil stabilisation in geotechnical and economic aspects.

#### 2.3.2. Process Control Optimization

The process control optimisation examines how much binder must be added based on the properties of the base material, e.g., water ratio, organic carbon content or impurity content, the process parameters of soil. In the experimental planning, this optimisation has been carried out using the factorial experiment. Figure 5a illustrates a simple $2^{2}$-factor experiment where two factors are tested at the two levels of various binders $2^{2}$. This method is used to evaluate how two different factors affect the final result. The factor test can be performed with a higher resolution, e.g., by increasing the number of levels to three when the $2^{3}$-factor is obtained. Similar to the simplex method, higher resolution means that the number of trial experiments increases. The choice of the resolution basically depends on the real project situation (amount, type and quality of soil that should be stabilised) and which interactions are expected to occur in soil treatment [69].

#### 2.3.3. Total Binder Optimization

In a total optimisation stage, simplex experimental trials from the mixture optimisation are combined in a factorial trials with process variables. In this case, the combined experimental set-up then becomes advanced, as illustrated schematically in Figure 5b. The advantage of this approach is that we find an optimum value both in terms of binder combination and binder quantity, based on the important process parameters. Otherwise, if we only investigate the optimum of stabilising mixture, based on only binder combination, we may miss the unforeseen and unwanted influence from the external process parameters. The reverse applies if we only optimise binder amount based on the process parameters.

This type of the total optimisation gives us a robust basis for being able to vary both the binder combinations and the amount of binder in a project, without risking either final quality or project economics. As a result, the combined methodology adopted from [70] provides both optimal binder mix and optimal binder quantity in relation to the process parameters.

## 3. Results and Discussion

The dynamics of the P-wave velocity, corresponding to soil strength as a function of curing time is shown in Figure 6. For the interpretability reasons, only the values of the soil specimens stabilised by one type of binder are shown. For soils that are sandy, added energy fly ash and bio fly ash at all ages of curing the sandy soil exhibited almost no effects on the compressive strength (brown lines in Figure 6) with very low values of P-waves (<250 m/s). Moreover, the dynamics remain stable for this case and do not change significantly for the period of 90 days. In contrast, stabilisation of soil by bio fly ash shows a clearly visible sharp increase in the P-wave velocity starting from day 28 from 350 m/s up to 3000 and 3100 for the two different types of bio flay ash (blue line in Figure 6).

Strong effects from added OPC on strength gain at an early age on the specimen are notable. The reaction of the two types of cement with different soil specimens varied, but the general trend is comparative: the values of P-waves start from 2400 and 265 m/s for the two types of cement and gradually continue to gain in strength until the P-waves are recorded at speed of 2600 and 3000 m/s, respectively. With sandy soil, the reaction of Portland cement type Cem II/A-V with the sandy fraction is comparable for both types. The gap in the P-wave speed for lime (purple line in Figure 6) and bio fly ash (green line in Figure 6) between 28 and 90 days is caused by the disturbances during cutting and demolding process of the specimens.

This disturbance has not affected tested specimens with high strength (cement and slag), i.e., the bond between the particles of the specimens stabilised with lime, energy ash and bio fly ash was so small that they broke during handling. Furthermore, lime, energy ash and bio fly ash have actively bounded the aggregates and for the small elongation levels that arise in the connection with seismic measurements, the bonds remain intact because of the larger elongation levels that arise during cutting and demolding, so the bonds are damaged. The strength of soil stabilised by lime reached its peak on day 28th when values of the P-waves were recorded as 950 and 1250 m/s for the two sets of measurements. Afterwards, the gain in strength decreased and values of P-waves dropped until 500 m/s.

The velocity of the P-waves propagating specimens stabilised by binary binders are shown in Figure 7. Specifically, it shows samples stabilised by cement (as a reference), slag and blended binary mixtures in combinations for binders as slag/lime and lime/slag with proportions: 0.67/0.33 for both combinations. A comparison between the slag and binary binders containing lime and slag shows a faster curing process when slag is combined with lime. Figure 7 furthermore shows that the two binary binders lime/slag and slag/lime have equally high or higher P-wave velocity on day 90 as cement. This can be compared with the values of the compressive strength reported for the day 90th for the strength measurements of soil stabilised by cement, lime and GGBFS (Tables 7 and 8). Thus, from Table 7 we can see that in terms of strength, cement and slag/lime in proportions 0.67/0.33 are equivalent, because soil stabilised by binary binders lime/slag (0.67/0.33) has a clearly lower strength. This phenomenon may have been owing to the nonlinearities of the stabilised soil material, i.e., elongation-dependent stiffness.

The results from the measured P-wave velocities indicating strength gain (UCS) are reported in Tables 1–8 for days 7, 14, 28 and 90, as well as the remaining values for the same days, respectively. The analysis of Table 1, which shows measured values of P-waves on the day 7th of curing period of stabilised soil, gives the following remarks. The presence of cement and energy fly ash (combination AD) influences the stabilisation characteristics of a soil with P-waves (coefficient 2408.8) by T = 47 °C. Likewise, the addition of lime and slag (combination BC) into soil results in the increase of P-wave coefficient up to 5474.4. The compound of lime and slag acts as an accelerator for soil hardening and in their presence increase the strength gain.

Correspondingly, adding slag (GGBFS) and bio fly ash (combination CE) leads to the increase of bonding soil particles, which is also important in providing higher particle friction and better packing. As a result, soil strength increases which is reflected in the higher speed of the elastic waves (P-wave coefficient = 2453.1). The measurements were done by T = 47 °C for all the data in Table 1. Depending on the amount of added slag (type Merit 5000) and Portland cement, bonding soil particles generally increases the UCS values compared to the one in the fly ashes, which demonstrated poor or low effects. The improvement is introduced by glueing of soil particles with cementitious binder at the points of contact by stabilising agents.

Similarly, the combination AE (cement and bio flay ash) also increased the soil strength, as the stabiliser content was increased in specimens. This results in the P-wave coefficient reached 5648.6 m/s. As for triple blends, upon exposure to added slag GGBFS and cement into binder blends, the compressive strength of a stabilised soil is gradually increased after the day 28th. This proves the important role of cement and lime for stabilisation. Thus, the rate of increase may vary for blends ACD—cement/slag/fly ash (P-wave coefficient = 13,776.3 m/s), ABE–cement/lime/bio fly ash (P-wave coefficient = 5768.0), combination ABD—cement/lime/energy fly ash (P-wave coefficient = 4591.6). Mechanically, a sandy soil system, when stabilised with either Portland cement, GGBFS or lime, provides improved shear and compressive strength, expressed in the UCS values, as it is also reflected in the P-waves velocity. Table 2 shows the remaining notable effects on day 7th of curing period after backward elimination. Here the combination AE (cement/bio fly ash) demonstrated the highest values of P-wave coefficient (5213.9) followed by the blend BC (lime/slag) with the P-wave coefficient of 4311.8.

The analysis of Table 3 demonstrates measured values of P-waves on the day 14th of curing period. Here we can note that the addition of single binders contributed to the increase of P-wave coefficient, which is attributed to the development of its strength. Specifically, added Portland cement resulted in the P-waves coefficient raising up to 2645.6, followed by bio fly ash (P-wave coefficient = 1476.8) and lime (P-wave coefficient = 781.8), while the effects from the energy fly ash and GGBFS are similar (211.8 and 208.5, respectively). In contrast, blended binders have a more notable effects on the formation of bonded particles, which strengthen a soil that is stabilised with the cement/slag GGBFS (blend AC, P-wave = 5229.2), lime/GGBFS (blend BC, P-wave = 7920.3) and slag/energy fly ash (blend CE, P-waves = 4819.4). As a result, this produces a higher strength, which is mirrored in the measured speed of the elastic P-waves.

Table 4 shows the remaining effects on day 14th after the backward elimination.

We observe here that Portland cement and lime have different effects on stabilisation (compare P-wave coefficient for cement P-wave = 2572.1 against P-wave = 872.6 for lime), since they differ in their chemical nature, although both provide calcium, which is necessary for soil hardening. A notably high value is shown by bio fly ash (P-wave = 1192.2) due to its mode of reaction with particles. Remarkably high values are also demonstrated by the combinations AC (cement/GGBFS, P-wave = 4811.3), AD (cement/energy fly ash P-wave = 3085.3), BC (lime/slag P-wave = 6901.9) and CE (GGBFS/bio fly ash, P-wave = 4756.8). This well demonstrates that reaction products may eventually differ in real cases of soil stabilisation by blended binders. Moreover, certain triple combinations demonstrated negative values, such as, for instance, ADE (Portland cement/energy fly ash/bio fly ash, P-waves = −29,959.3). Clearly, adding cement as a stabilising agent is beneficial for soil, as it produces strength-developing hydration product in a soil mixture through bonding with particles, which finally leads to the improved strength of soil.

The inspection of Table 5, showing the results showing P-wave velocities on the day 28th of curing period by temperature 46 °C, produces following remarks. The difference in values of the P-wave velocities, compared to the previous Tables on days 7th and 14th implies that the time of the reaction plays an important role on the stabilisation in soils, which generally results from the chemical processes caused by the stabilising agent as binder or the blended mixture of double of triple binders. These factors include the two different processes. The flocculation effects come to the effect immediate upon the contact of binder with soil particles.

On the contrary, pozzolanic and hydration effects are generally time-dependent. Therefore, we can see the differences in stabilisation, as reflected by the P-waves speed on days 7th, 14th and 28th by the same stabilising agents. For instance, the combinations of cement/lime (blend AB) or cement/GGBSF (blend AC) for the day 28th, which give the P-wave coefficients as 928.1 and 5917.3 against similar values on day 14th and 7th. Thus, P-wave coefficients for the combination AB for day 14th: 828.6 and for the day 7: 730.2, respectively. Likewise, for the combination AC P-waves are recorded as 1345.3 for day 7th and 5229.2 for day 14th. This can be explained by the inherent nature of the Portland cement, in addition to the added binders, which contributes to the increase of strength and plasticity reduction over the time of curing.

Table 6 shows the remaining significant effects on soil stabilisation from different binders on day 28th after the backward elimination in the regression analysis. The dynamics in P-waves which generally increases over time is explained by the increased soil strength as a result of the stabilisation, where plasticity of soil is reduced and particles are bonded to each other in a soil-blend mixture. Thus, the compressive strength and load-bearing properties of soil are improved, which results in the increased stiffness and compaction of soil and reduced porosity. This corresponds to the higher of the P-wave velocity which rises in a more dense structure, due to the chemical processes in soil caused by binders.

Table 7 shows that slag is the main factor that contributes most to the compressive strength among the other binders followed by the OPC, which comes second in terms of strength. The combination of slag and energy ash has the least impact on the compressive strength of soil. Furthermore, Table 7 shows linear effects between cement, lime and slag as factors affecting soil stabilisation and gain of strength when comparing the values of P-waves, corresponding to the gain of strength over curing time of 3 months. By closer examination of Table 7, the effects from lime and slag that are reflected in seismic measurements (7th, 14th and 28th daily values in Table 1, Table 3 and Table 5, respectively) cannot be confirmed for the values of strength recorded on day 90. Here high values are demonstrated by the blend lime/slag (BC, P-wave = 6004.0), lime/energy fly slag (BD, P-waves = 4620.5) and cement/slag (AC, 5147.8). Table 8 shows the remaining significant effects on day 90 after backward elimination in regression analysis.

The combination of lime and slag has a significant effect on hardening process of soil, but no effects on final strength development. These observations also confirm previous study [68] on the three different types of the fine-grained moraines, of which the two samples with the largest clay content demonstrated a significant interaction between lime and slag, while the third sample with the lowest clay content shown the same pattern, as shown in Table 7. This indicates that bearing gravel used in the study is inert and therefore well suited to study the effects of different binders and their interactions. The remaining significant effects on day 90th after backward elimination show notable values for single binders slag (P-wave = 8499.5) and cement (P-wave = 6042.5), as well as triple combination of blended mixture of binders Portland cement/ACD/slag GGBFS/fly ash (P-wave = 60,200.0), as shown in Table 8.

## 4. Conclusions

In this study, we demonstrated that blended binders perform better on soil stabilisation, as compared to the single binders. The experiments for optimal selection of binder combinations and amounts reduce the financial risks of the project, which are possible if soil is not sufficiently stabilised, and minimise the environmental impact through the reduced amounts of cement by replacement from other binders. Because the costs of the laboratory tests are expensive, optimising blended binders is essential for high performance of soil in real construction projects. In turn, the decisions based on the results of soil stabilisation and evaluated soil strength support solutions for technical, economic and environmental processes. Trial planning also provides the support for evaluation if optimal mixture is within a wide or narrow mixture range of blended binders. If it is narrow, then greater precision is required in the decisions, compared to the cases with a wide range. In contrast, if the range of binder choices is wide, other parameters can be decisive for final solutions. These includes, for instance, geotechnical properties of soil, economics or delivery security.

Stabilisation of poor subgrade soil intended for road construction is aimed at improving its engineering properties, commonly estimated as strength and stiffness. The growth of strength and stiffness is strongly dependent on time of curing and the effects from the stabilising agents which can be used both as single binders and as blended mixtures. Using mixed binders as stabilising agents may have diverse effects on the properties of soil. Therefore, the experimental testing of the effects of blenders is required. In this study, the comparative analysis was performed to assess the functional behavior of individual and blended (binary and triple) binders and curing time of 90 days using P-wave velocity. The development in P-wave velocity was compared both for individual and blended binders over the complete period of curing. The results indicate that strength growth in stabilised soil samples is nonlinear in both time and content of binders.

The analysis of the effects from various agents demonstrated that GGBFS contributes the most to the compressive strength development, followed by OPC. Moreover, we noted that blended mixtures perform better than single binders. Thus, the combination of lime and slag has a notable effect on soil hardening and final strength development. The combination of slag and energy fly ash contributes to the gain of strength, but has the least impact on compared to other blends. Factor test combining blended binders of various mixtures, enabled to investigate, under which circumstances two factors interact during the process of soil stabilisation, so that the quality of the final product to be approved or not, e.g., if soil treatment takes place in the sub-area with a high contaminant content and high water content. Tailored prevention strategies in road construction can be developed based on this procedure, by selectively increasing the amount of binders in the areas that constitute to the risk zones or regulating the permitted water ratio in the soil base material. From the geotechnical and engineering perspectives in context of road or highway construction, soil stabilisation should ideally have a lifespan of 80 years or more. From an environmental perspective, in case of contaminated soil, stabilisation of specimens should have a risk-reducing effect over the significantly longer periods of time.

In order to ensure that final product meets the approved requirements, both geotechnical construction of roads and physiochemical improvements of soil quality should be considered. At the same time, base material exhibits varying properties within the treatment surface, therefore the robustness of the binder mixture should be tested by factor test, as demonstrated in this study. Using simplex design method of binder combinations we estimated and compared the effects from blended mixtures and single binders (OPC, lime, GGBFS type Merit 5000, energy fly ash and bio flay ash) on soil stabilisation.

The robustness of soil stabilisation means that final binder mixture is able to provide the approved geotechnical and environmental quality of soil intended for road construction. This is necessary both for the short-term and long-term period of curing under all the conditions prevailing in soil treatment. The procedure of stabilisation can lead to the unnecessarily high binder costs and increase climate emissions through the binder’s production chain (especially for OPC) or the unnecessarily high technical performance given the basic requirements that govern the quality of the final product. In the traditional tests that consider only one factor at a time, the robustness of blending binder is often handled by the increased amount of binder, so that the results of the provided experiment ensure the avoided uncertainties that may exist: the individual physiochemical properties of soil and local environmental conditions of real project, e.g., specifics of roads and local climate. In this paper, we designed adaptation algorithms for selecting optimal binder combinations using simplex design combinatorics to improve soil properties and geotechnical parameters through stabilisation for industrial projects of road and highway construction.

## Figures and Tables

**Figure 1 materials-15-07798-f001:**
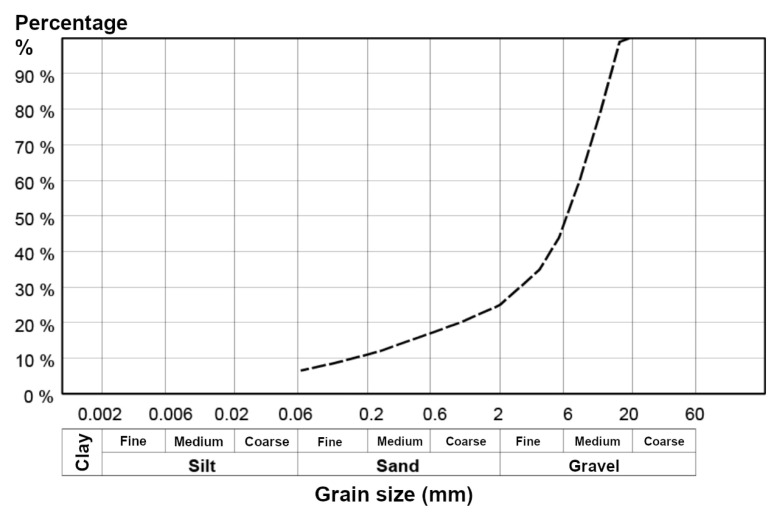
Grain size distribution for soil used in this study.

**Figure 2 materials-15-07798-f002:**
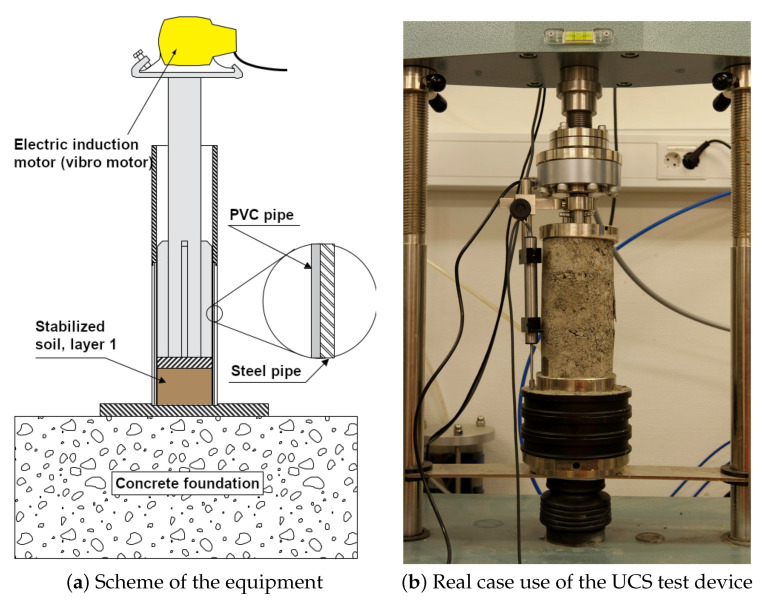
Compaction equipment for soil stabilisation (**a**); Specimen that reached failure (**b**).

**Figure 3 materials-15-07798-f003:**
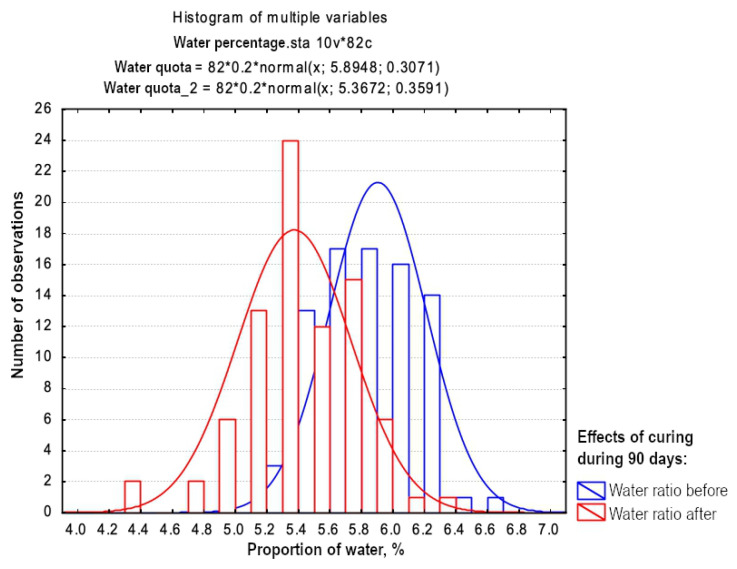
Histogram of water content in soil specimens.

**Figure 4 materials-15-07798-f004:**
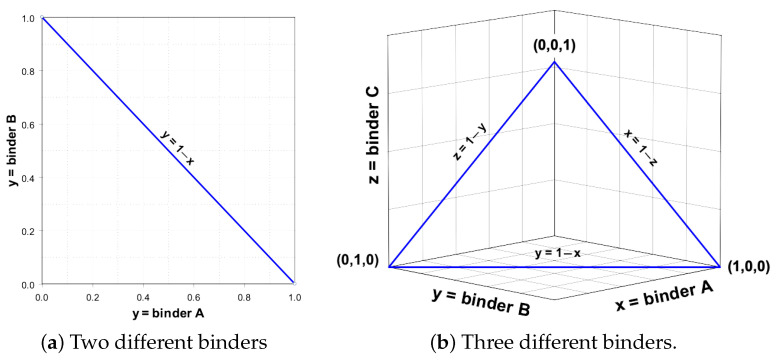
Application of simplex lattice design showing possible binder combinations.

**Figure 5 materials-15-07798-f005:**
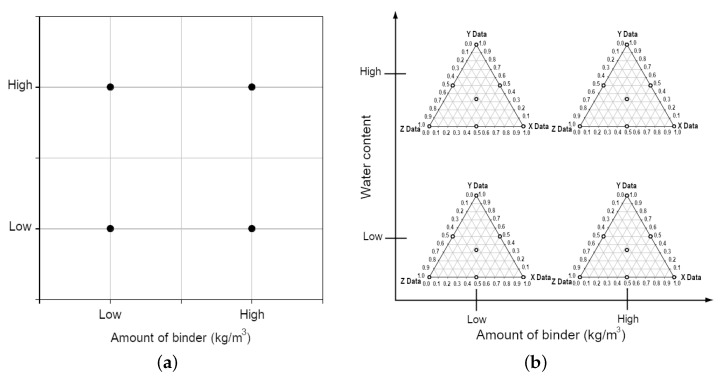
An example of a $2^{2}$-factor experiment with binder quantity and water ratio as factors (**a**); An example of a $2^{2}$-factor trial in combination with simplex trials (**b**).

**Figure 6 materials-15-07798-f006:**
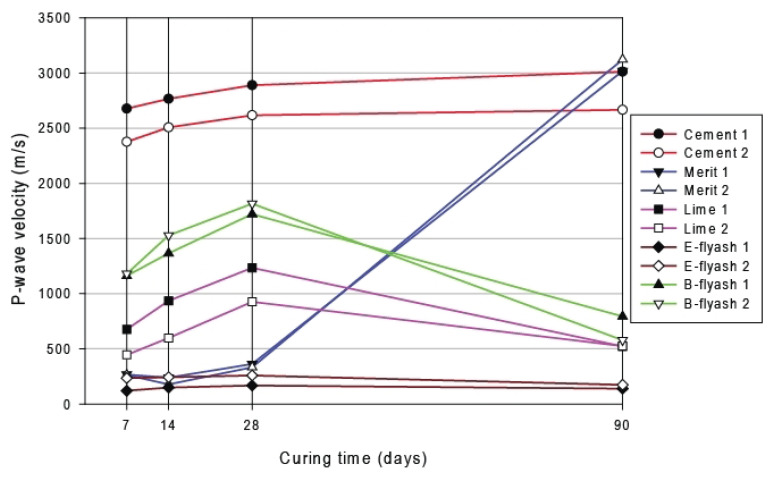
P-wave velocity as a function of curing time for individual binders.

**Figure 7 materials-15-07798-f007:**
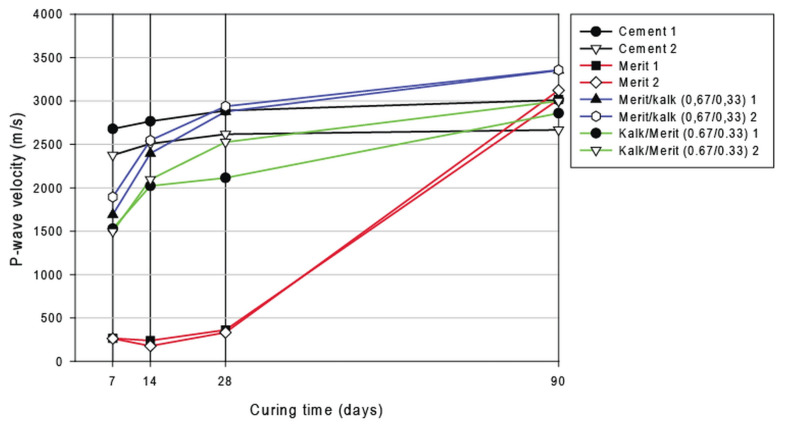
P-wave velocity as a function of curing time for individual and blended binary binders.

**Table 1 materials-15-07798-t001:** The effects from various binders on soil stabilisation on day 7th of curing time, measured by P-wave velocities.

Factors	P-Wave Coeff.	Std. Err.	T = 47 °C	*p*	−95.% Cnf. Limt	+95.% Cnf. Limt
(A) Cement	2538.3	217.223	11.68544	0.000000	2101.4	2975.35
(B) Lime	563.5	217.223	2.59394	0.012613	126.5	1000.46
(C) GGBFS	277.5	217.223	1.27759	0.207668	−159.5	714.52
(D) E_Fly ash	174.4	217.223	0.80280	0.426134	−262.6	611.38
(E) B_Fly ash	1192.9	217.223	5.49141	0.000002	755.9	1629.86
AB	730.2	972.388	0.75095	0.456429	−1226.0	2686.40
AC	1345.3	972.388	1.38346	0.173061	−610.9	3301.45
AD	2408.8	972.388	2.47718	0.016896	452.6	4364.97
AE	−1599.1	972.388	−1.64453	0.106740	−3555.3	357.07
BC	5474.4	972.388	5.62986	0.000001	3518.2	7430.60
BD	108.9	972.388	0.11203	0.911278	−1847.3	2065.13
BE	−962.7	972.388	−0.99003	0.327230	−2918.9	993.50
CD	−18.4	972.388	−0.01892	0.984982	−1974.6	1937.79
CE	2453.1	972.388	2.52278	0.015086	496.9	4409.31
DE	−201.2	972.388	−0.20692	0.836967	−2157.4	1754.99
ABC	−7209.2	6848.162	−1.05272	0.297852	−20,985.9	6567.54
ABD	4591.6	6848.162	0.67049	0.505827	−9185.1	18,368.35
ABE	5768.0	6848.162	0.84227	0.403902	−8008.7	19,544.73
ACD	13,776.3	6848.162	2.01168	0.050007	−0.4	27,553.01
ACE	5997.5	6848.162	0.87578	0.385601	−7779.2	19,774.24
ADE	−23,505.9	6848.162	−3.43244	0.001257	−37,282.6	−9729.20
BCD	−3246.1	6848.162	−0.47402	0.637683	−17,022.9	10,530.59
BCE	−13,506.6	6848.162	−1.97230	0.054476	−27,283.3	270.12
BDE	5566.1	6848.162	0.81279	0.420437	−8210.6	19,342.84
CDE	407.6	6848.162	0.05952	0.952793	−13,369.1	14,184.31
AB(A-B)	1814.7	2064.817	0.87886	0.383951	−2339.2	5968.55
AC(A-C)	1048.7	2064.817	0.50789	0.613907	−3105.2	5202.57
AD(A-D)	−3526.9	2064.817	−1.70809	0.094218	−7680.8	626.99
AE(A-E)	5648.6	2064.817	2.73563	0.008756	1494.7	9802.45
BC(B-C)	−2271.6	2064.817	−1.10012	0.276881	−6425.4	1882.32
BD(B-D)	−265.0	2064.817	−0.12835	0.898421	−4418.9	3888.86
BE(B-E)	3986.1	2064.817	1.93050	0.059591	−167.7	8140.01
CD(-D)	−277.3	2064.817	−0.13429	0.893750	−4431.2	3876.60
CE(C-E)	75.9	2064.817	0.03677	0.970822	−4077.9	4229.81
DE(D-E)	528.2	2064.817	0.25583	0.799196	−3625.6	4682.13

Factors are the combinations of binders. All the factors are included in the cubic model. The color-marked rows indicate a significance level of values below the p = 0.05. The coefficients for recoded computations are as follows: Var.: Vp_rod_7; R-sqr = 0.8806; Adj: 0.7942 (Resultat_rev091123.sta). Five-factor mixture design; Mixture total = 1, 82 runs. DV: Vp_rod_7; MS Residual = 95,430.29. Std. Err. is a standard error (standard deviation of sample distribution).

**Table 2 materials-15-07798-t002:** Remaining significant effects on day 7th after the backward elimination.

Factors	P-Wave Coeff.	Std. Err.	t (72)	*p*	−95.% Cnf. Limt	+95.% Cnf. Limt
(A) Cement	2643.3	138.311	19.11103	0.000000	2367.6	2919.0
(B) Lime	601.7	133.783	4.49725	0.000026	335.0	868.4
(C) GGBFS	497.6	152.147	3.27048	0.001649	194.3	800.9
(D) E_Fly ash	272.1	134.969	2.01612	0.047519	3.1	541.2
(E) B_Fly ash	893.4	139.921	6.38482	0.000000	614.4	1172.3
AD	2789.3	809.587	3.44535	0.000955	1175.4	4403.2
BC	4311.8	788.740	5.46665	0.000001	2739.4	5884.1
CE	2368.8	790.947	2.99492	0.003762	792.1	3945.5
ADE	−27,704.7	6364.530	−4.35298	0.000044	−40,392.1	−15,017.2
AE(A-E)	5213.9	2104.584	2.47741	0.015582	1018.5	9409.3

A significance level below the p = 0.05 is noted for all the recorded values in Table 2. Annotation for Table 2: Coeffs (recoded comps); Var.: Vp_rod_7; R-sqr = 0.8014; Adj: 0.7765 (Resultat_rev091123.sta). Five-factor mixture design; Mixture total = 1, 82 Runs DV: Vp_rod_7; MS Residual = 103,610.4. Factors stand for combination of binders.

**Table 3 materials-15-07798-t003:** The effects from various binders on soil stabilisation on day 14th of curing time, measured by P-wave velocities.

Factors	P-Wave Coeff.	Std. Err.	t(47)	*p*	−95.% Cnf. Limt	+95.% Cnf. Limt
(A) Cement	2645.6	202.397	13.07116	0.000000	2238.4	3052.7
(B) Lime	781.8	202.397	3.86294	0.000342	374.7	1189.0
(C) GGBFS	208.5	202.397	1.03027	0.308156	−198.6	615.7
(D) E_Fly ash	211.8	202.397	1.04639	0.300731	−195.4	619.0
(E) B_Fly ash	1476.8	202.397	7.29644	0.000000	1069.6	1883.9
AB	828.6	906.019	0.91455	0.365095	−994.1	2651.3
AC	5229.2	906.019	5.77161	0.000001	3406.5	7051.9
AD	2644.7	906.019	2.91908	0.005374	822.1	4467.4
AE	−1669.3	906.019	−1.84248	0.071717	−3492.0	153.4
BC	7920.3	906.019	8.74183	0.000000	6097.6	9742.9
BD	351.7	906.019	0.38813	0.699673	−1471.0	2174.3
BE	−980.5	906.019	−1.08217	0.284699	−2803.1	842.2
CD	238.4	906.019	0.26314	0.793591	−1584.3	2061.1
CE	4819.4	906.019	5.31932	0.000003	2996.7	6642.1
DE	−239.9	906.019	−0.26481	0.792311	−2062.6	1582.7
ABC	−16,343.2	6380.749	−2.56133	0.013696	−29,179.6	−3506.8
ABD	4655.7	6380.749	0.72965	0.469228	−8180.7	17,492.1
ABE	7752.7	6380.749	1.21501	0.230431	−5083.7	20,589.1
ACD	10,057.6	6380.749	1.57624	0.121677	−2778.8	22,894.0
ACE	−3180.4	6380.749	−0.49843	0.620506	−16,016.8	9656.1
ADE	−25542.1	6380.749	−4.00299	0.000221	−38,378.5	−12,705.7
BCD	5030.5	6380.749	0.78839	0.434430	−7805.9	17,866.9
BCE	−8617.9	6380.749	−1.35061	0.183289	−21,454.3	4218.5
BDE	6423.5	6380.749	1.00670	0.319233	−6412.9	19,259.9
CDE	3468.8	6380.749	0.54363	0.589265	−9367.6	16,305.2
AB(A-B)	1873.4	1923.886	0.97375	0.335164	−1997.0	5743.7
AC(A-C)	−7129.4	1923.886	−3.70572	0.000555	−10,999.7	−3259.0
AD(A-D)	−3098.9	1923.886	−1.61077	0.113926	−6969.3	771.4
AE(A-E)	6913.4	1923.886	3.59347	0.000779	3043.1	10,783.8
BC(B-C)	−4438.2	1923.886	−2.30690	0.025515	−8308.6	−567.8
BD(B-D)	59.4	1923.886	0.03089	0.975491	−3810.9	3929.8
BE(B-E)	4666.2	1923.886	2.42541	0.019186	795.9	8536.6
CD(C-D)	472.9	1923.886	0.24582	0.806890	−3397.4	4343.3
CE(C-E)	2542.3	1923.886	1.32143	0.192754	−1328.1	6412.6
DE(D-E)	818.4	1923.886	0.42541	0.672482	−3051.9	4688.8

Factors indicate combination of binders. All the factors included in the cubic model. The color-marked rows indicate a significance level of values below the p = 0.05. Coeffs (recoded comps); Var.:Vp_rod_14; R-sqr = 0.9207; Adj: 0.8633 (Resultat_rev091123.sta). Five-factor mixture design; Mixture total = 1, 82 Runs. DV: Vp_rod_14; MS Residual = 82,847.91. Std. Err. is a standard error (standard deviation of sample distribution).

**Table 4 materials-15-07798-t004:** Remaining significant effects on day 14 after the backward elimination.

Factors	P-Wave Coeff.	Std. Err.	t(68)	*p*	−95.% Cnf. Limt	+95.% Cnf. Limt
(A) Cement	2572.1	149.090	17.25165	0.000000	2274.6	2869.6
(B) Lime	872.6	131.407	6.64061	0.000000	610.4	1134.8
(C) GGBFS	362.3	168.836	2.14575	0.035465	25.4	699.2
(D) E_Fly ash	385.0	127.497	3.01993	0.003559	130.6	639.4
(E) B_Fly ash	1192.2	134.910	8.83687	0.000000	923.0	1461.4
AC	4811.3	754.640	6.37562	0.000000	3305.4	6317.2
AD	3085.3	768.942	4.01236	0.000152	1550.9	4619.7
BC	6901.9	753.201	9.16339	0.000000	5398.9	8404.9
CE	4756.8	756.913	6.28445	0.000000	3246.4	6267.2
ADE	−29,959.3	5994.247	−4.99801	0.000004	−41,920.6	−17,998.0
AC(A-C)	−6802.8	1990.624	−3.41743	0.001071	−10,775.0	−2830.6
AE(A-E)	6630.1	1988.645	3.33396	0.001389	2661.8	10,598.3
BC(B-C)	−4518.4	1984.089	−2.27734	0.025916	−8477.6	−559.3
BE(B-E)	3976.0	1981.513	2.00657	0.048774	22.0	7930.1

A significance level below the p = 0.05 is noted for all the recorded values in Table 4. Coeffs (recoded comps); Var.:Vp_rod_14; R-sqr = 0.8728; Adj: 0.8484 (Resultat_rev091123.sta). Five-factor mixture design; Mixture total = 1, 82 Runs. DV: Vp_rod_14; MS Residual = 91,875.45. Factors are the combinations of binders.

**Table 5 materials-15-07798-t005:** The effects from various binders on soil stabilisation on day 28th of curing, measured by P-wave velocities.

Factors	P-Wave Coeff.	Std. Err.	t(47)	*p*	−95.% Cnf. Limt	+95.% Cnf. Limt
(A) Cement	2757.6	227.599	12.11585	0.000000	2299.7	3215.4
(B) Lime	1094.6	227.599	4.80915	0.000016	636.7	1552.4
(C) GGBFS	353.0	227.599	1.55109	0.127589	−104.8	810.9
(D) E_Fly ash	213.6	227.599	0.93831	0.352883	−244.3	671.4
(E) B_Fly ash	1798.2	227.599	7.90068	0.000000	1340.3	2256.1
AB	928.1	1018.836	0.91097	0.366960	−1121.5	2977.8
AC	5917.3	1018.836	5.80793	0.000001	3867.7	7967.0
AD	2926.6	1018.836	2.87252	0.006093	877.0	4976.3
AE	−1973.4	1018.836	−1.93691	0.058781	−4023.0	76.2
BC	8789.8	1018.836	8.62726	0.000000	6740.1	10,839.4
BD	1194.6	1018.836	1.17251	0.246903	−855.0	3244.2
BE	−1261.8	1018.836	−1.23850	0.221680	−3311.5	787.8
CD	122.4	1018.836	0.12018	0.904855	−1927.2	2172.1
CE	6434.7	1018.836	6.31573	0.000000	4385.1	8484.3
DE	388.6	1018.836	0.38137	0.704650	−1661.1	2438.2
ABC	−14,519.3	7175.280	−2.02352	0.048727	−28,954.1	−84.5
ABD	3311.1	7175.280	0.46146	0.646600	−11,123.7	17,745.9
ABE	9603.0	7175.280	1.33834	0.187224	−4831.8	24,037.8
ACD	13,143.3	7175.280	1.83174	0.073332	−1291.5	27,578.1
ACE	−6703.8	7175.280	−0.93429	0.354928	−21,138.6	7731.0
ADE	−26,777.2	7175.280	−3.73187	0.000512	−41,212.0	−12,342.4
BCD	12,947.4	7175.280	1.80444	0.077572	−1487.4	27,382.2
BCE	−11,210.9	7175.280	−1.56243	0.124896	−25,645.7	3223.9
BDE	6493.1	7175.280	0.90493	0.370118	−7941.7	20,927.9
CDE	9002.6	7175.280	1.25466	0.215805	−5432.2	23,437.4
AB(A-B)	1405.5	2163.448	0.64968	0.519063	−2946.8	5757.8
AC(A-C)	−7717.7	2163.448	−3.56730	0.000843	−12,070.0	−3365.4
AD(A-D)	−3623.9	2163.448	−1.67507	0.100562	−7976.2	728.4
AE(A-E)	7851.3	2163.448	3.62908	0.000700	3499.0	12,203.6
BC(B-C)	−4806.3	2163.448	−2.22158	0.031165	−9158.6	−454.0
BD(B-D)	−678.2	2163.448	−0.31346	0.755320	−5030.4	3674.1
BE(B-E)	4889.2	2163.448	2.25992	0.028501	536.9	9241.5
CD(C-D)	−202.8	2163.448	−0.09372	0.925729	−4555.1	4149.5
CE(C-E)	6361.9	2163.448	2.94064	0.005068	2009.6	10,714.2
DE(D-E)	−1936.1	2163.448	−0.89490	0.375400	−6288.4	2416.2

Factors signify the combination of binders. All the factors included in the cubic model. The color-marked rows indicate a significance level of values below the p = 0.05. Coeffs (recoded comps); Var.:Vp_rod_28_f; R-sqr = 0.9152; Adj: 0.8538 (Resultat_rev091123.sta) Five-factor mixture design; Mixture total = 1, 82 Runs. DV: Vp_rod_28_f; MS Residual = 104,764.9. Std. Err. is a standard error (standard deviation of sample distribution).

**Table 6 materials-15-07798-t006:** Remaining significant effects on day 28th after the backward elimination.

Factors	P-Wave Coeff.	Std. Err.	t(69)	*p*	−95.% Cnf. Limt	+95.% Cnf. Limt
(A) Cement	2641.7	188.071	14.04626	0.000000	2266.5	3016.9
(B) Lime	1234.2	159.337	7.74602	0.000000	916.4	1552.1
(C) GGBFS	535.2	213.044	2.51231	0.014337	110.2	960.2
(D) E_Fly ash	559.9	160.875	3.48025	0.000873	238.9	880.8
(E) B_Fly ash	1504.3	169.277	8.88679	0.000000	1166.6	1842.0
AC	5570.6	951.996	5.85146	0.000000	3671.4	7469.8
AD	3190.2	970.228	3.28813	0.001589	1254.7	5125.8
BC	7892.0	951.349	8.29563	0.000000	5994.2	9789.9
CE	6318.7	950.481	6.64791	0.000000	4422.6	8214.9
ADE	−29,928.3	7562.989	−3.95721	0.000182	−45,016.1	−14,840.6
AC(A-C)	−7511.2	2511.772	−2.99042	0.003860	−12,522.1	−2500.4
AE(A-E)	7551.3	2509.255	3.00936	0.003653	2545.4	12,557.1
CE(C-E)	5855.4	2503.606	2.33880	0.022249	860.9	10,850.0

A significance level below the p = 0.05 is noted for all the recorded values in Table 6. Coeffs (recoded comps); Var.:Vp_rod_28_f; R-sqr = 0.8261; Adj: 0.7958 (Resultat_rev091123.sta). Five-factor mixture design; Mixture total = 1, 82 Runs. DV: Vp_rod_28_f; MS Residual = 146,270.9.

**Table 7 materials-15-07798-t007:** The effects from various binders on soil stabilisation on day 90th of curing time, measured by P-wave velocities.

Factors	P-Wave Coeff.	Std. Err.	t(47)	*p*	−95.% Cnf. Limt	+95.% Cnf. Limt
(A) Cement	5992.1	338.88	17.68189	0.000000	5310.3	6673.8
(B) Lime	1387.2	338.88	4.09355	0.000166	705.5	2069.0
(C) GGBFS	6910.7	338.88	20.39270	0.000000	6229.0	7592.4
(D) E_Fly ash	263.6	338.88	0.77791	0.440522	−418.1	945.4
(E) B_Fly ash	1577.6	338.88	4.65543	0.000027	895.9	2259.4
AB	−2448.3	1516.98	−1.61394	0.113235	−5500.1	603.5
AC	5147.8	1516.98	3.39344	0.001410	2096.0	8199.6
AD	1775.3	1516.98	1.17030	0.247780	−1276.4	4827.1
AE	1710.6	1516.98	1.12761	0.265208	−1341.2	4762.3
BC	6004.0	1516.98	3.95785	0.000254	2952.2	9055.8
BD	4620.5	1516.98	3.04586	0.003797	1568.7	7672.3
BE	834.3	1516.98	0.54994	0.584961	−2217.5	3886.0
CD	−12,683.0	1516.98	−8.36066	0.000000	−15,734.8	−9631.2
CE	6310.9	1516.98	4.16015	0.000134	3259.1	9362.7
DE	2361.3	1516.98	1.55657	0.126282	−690.5	5413.1
ABC	−14620.6	10,683.55	−1.36851	0.177658	−36,113.1	6872.0
ABD	−1054.7	10,683.55	−0.09872	0.921779	−22,547.2	20,437.8
ABE	−6205.2	10,683.55	−0.58082	0.564140	−27,697.7	15,287.4
ACD	45,351.2	10,683.55	4.24496	0.000102	23,858.7	66,843.7
ACE	−45,649.2	10,683.55	−4.27285	0.000093	−67,141.7	−24,156.6
ADE	10,067.1	10,683.55	0.94230	0.350859	−11,425.4	31,559.6
BCD	54,712.7	10,683.55	5.12122	0.000006	33,220.2	76,205.3
BCE	−21,262.1	10,683.55	−1.99017	0.052406	−42,754.6	230.4
BDE	7008.6	10,683.55	0.65602	0.515009	−14,483.9	28,501.2
CDE	43,775.5	10,683.55	4.09747	0.000164	22,283.0	65,268.0
AB(A-B)	1876.0	3221.24	0.58237	0.563099	−4604.3	8356.3
AC(A-C)	−6009.6	3221.24	−1.86562	0.068343	−12,489.9	470.7
AD(A-D)	648.5	3221.24	0.20132	0.841319	−5831.8	7128.8
AE(A-E)	5207.7	3221.24	1.61667	0.112643	−1272.6	11,688.0
BC(B-C)	−8918.3	3221.24	−2.76860	0.008031	−15,398.6	−2438.0
BD(B-D)	419.5	3221.24	0.13024	0.896934	−6060.8	6899.8
BE(B-E)	−887.0	3221.24	−0.27537	0.784238	−7367.3	5593.3
CD(C-D)	−10,632.0	3221.24	−3.30060	0.001847	−17,112.3	−4151.7
CE(C-E)	7909.8	3221.24	2.45551	0.017822	1429.5	14,390.1
DE(D-E)	−3409.2	3221.24	−1.05835	0.295307	−9889.5	3071.1

All the factors included in the cubic model. The color-marked values idicate a significance level below p = 0.05. Coeffs (recoded comps); Var.:UCS; R-sqr = 0.9678; Adj: 0.9445 (Resultat_rev091123.sta). Five-factor mixture design; Mixture total = 1, 82 Runs. DV: UCS; MS Residual = 232,257.1 Std. Err. is a standard error (standard deviation of sample distribution).

**Table 8 materials-15-07798-t008:** Remaining significant effects on day 90th after backward elimination.

Factors	P-Wave Coeff.	Std. Err.	t(71)	*p*	−95.% Cnf. Limt	+95.% Cnf. Limt
(A) Cement	6042.5	238.21	25.3667	0.000000	5567.5	6517.5
(B) Lime	1497.8	235.56	6.3587	0.000000	1028.1	1967.5
(C) GGBFS	8499.5	269.19	31.5738	0.000000	7962.7	9036.3
(D) E_Fly ash	1006.3	265.81	3.7857	0.000317	476.3	1536.3
(E) B_Fly ash	1985.8	238.21	8.3363	0.000000	1510.8	2460.7
CD	−17,791.7	1675.96	−10.6158	0.000000	−21,133.5	−14,449.9
ACD	60,200.0	12,482.71	4.8227	0.000008	35,310.2	85,089.8
ACE	−24,816.7	12,149.73	−2.0426	0.044809	−49,042.6	−590.9
BCD	78,876.2	12,501.89	6.3091	0.000000	53,948.1	103,804.2
CDE	59,769.4	12,482.71	4.7882	0.000009	34,879.6	84,659.2
CD(C-D)	−11,168.5	4107.77	−2.7189	0.008227	−19,359.2	−2977.9

A significance level p = 0.05 is noted for all the recorded values. Coeffs (recoded comps); Var.:UCS; R-sqr = 0.9164; Adj: 0.9046 (Resultat_rev091123.sta). Five-factor mixture design; Mixture total = 1, 82 Runs. DV: UCS; MS Residual = 39,9436.4.

## Data Availability

Not applicable.

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
