# Peer review of "Dynamics of Strength Gain in Sandy Soil Stabilised with Mixed Binders Evaluated by Elastic P-Waves during Compressive Loading"

_materials, 2022, doi:10.3390/ma15217798_

Round 1

Reviewer 1 Report

Article is of high scientific rank. The topic of the publication is extremely relevant in connection with stabilization of poor subgrade soil for improving its engineering properties and stiffness. For this purpose, single binders and blended mixtures of 5 binding components are used during 90 days: strength test for the Unconfined Compressive Strength and seismic tests on the stabilized samples. The publication correlates with building materials and agro-ecology, which means an interdisciplinary nature.
Small technical gaps such as the lack of a unit on the vertical scale in fig. 3, the names used in fig. 4a are not in English and should be replaced with their English names.
It is not clear how the values ​​in tables 1-8 are obtained for determining the effects from various binders on soil stabilization, measured by P-wave velocities for the different days of curing time. I recommend that this part of the work be explained in more detail so that it is clear to a wide range of readers, not only to narrow specialists in the field. At the same time, the conclusions are extensive and my recommendation is to optimize them.

Author Response

Dear Editors,

Please find attached the revised version of the paper. We have carefully followed the comments and critical suggestions and corrected the manuscript accordingly.

All the changes in the text are colored yellow, for the convenience of track changes.

We have proofread the paper throughout and corrected all occasional typographical errors, incorrect punctuation, spelling and minor grammar mistakes, typesetting misprints and inaccurate words where necessary. The replies to the comments in the reviewer’s reports are listed below. Using the opportunity, we thank the reviewer for careful reading of the paper which improved its first version.

With kind regards, - Authors (Per Lind and Polina Lemenkova).

30.10.2022.

Reviewer 1

No

Reviewer’s Comments

Author’s actions

1

Article is of high scientific rank. The topic of the publication is extremely relevant in connection with stabilization of poor subgrade soil for improving its engineering properties and stiffness. For this purpose, single binders and blended mixtures of 5 binding components are used during 90 days: strength test for the Unconfined Compressive Strength and seismic tests on the stabilized samples. The publication correlates with building materials and agro-ecology, which means an interdisciplinary nature.

General comment (no action is required).

2

Small technical gaps such as the lack of a unit on the vertical scale in fig. 3, the names used in fig. 4a are not in English and should be replaced with their English names.

Corrected: we replotted Fig. 3 (now Fig. 1 according to the updated numeration) and added a unit on the vertical scale (Percentage, %); Fig. 4a (now Fig. 2a) is replotted with names translated into English. The same is done for Fig. 5.

3

It is not clear how the values ​​in tables 1-8 are obtained for determining the effects from various binders on soil stabilization, measured by P-wave velocities for the different days of curing time.

I recommend that this part of the work be explained in more detail so that it is clear to a wide range of readers, not only to narrow specialists in the field. At the same time, the conclusions are extensive and my recommendation is to optimize them.

The P-wave coefficients in the tables 1-8 indicate the speed of P-waves. In general, the denser and more compacter the soil sample, the higher is the speed. The Factors show the combination of the binders which hare abbreviated initially in 5 major groups: A- cement, B – lime, C - slag GGBFS, D – energy fly ash, E – bio fly ash. Then these binders were combined with mixed combinations, e.g., AC = cement+ slag, AB = cement+lime; ABD = cement+ lime+ energy fly ash, etc, following this logic. These combinations are shown in the 1st column of the tables and indicate the cases where the P-wave-velocity was measured and the speed was evaluated. The other values indicate statistical values, e.g. the significance derived from p-value, standard error, residual, etc. So, these values show the statistical data processing for the case of samples stabilised by the combinations of these binders taken as single binders or as mixes of double or triple blends. As mentioned, higher P-wave velocities correspond to the better stabilization of soil which hardens due to the effects from binders. The conclusion section is modified, partially rewritten and shortened significantly, so that it should be better.

4

Are the conclusions supported by the results? – Can be improved.

Conclusions are rewritten, shortened and improved. Some phrases are modified, others are added for a better clearance.

Original review report:

Open Review

English language and style

( ) Extensive editing of English language and style required
( ) Moderate English changes required
( ) English language and style are fine/minor spell check required
(x) I don't feel qualified to judge about the English language and style

Yes

Can be improved

Must be improved

Not applicable

Does the introduction provide sufficient background and include all relevant references?

(x)

( )

( )

( )

Are all the cited references relevant to the research?

(x)

( )

( )

( )

Is the research design appropriate?

(x)

( )

( )

( )

Are the methods adequately described?

(x)

( )

( )

( )

Are the results clearly presented?

(x)

( )

( )

( )

Are the conclusions supported by the results?

( )

(x)

( )

( )

Comments and Suggestions for Authors

Article is of high scientific rank. The topic of the publication is extremely relevant in connection with stabilization of poor subgrade soil for improving its engineering properties and stiffness. For this purpose, single binders and blended mixtures of 5 binding components are used during 90 days: strength test for the Unconfined Compressive Strength and seismic tests on the stabilized samples. The publication correlates with building materials and agro-ecology, which means an interdisciplinary nature.
Small technical gaps such as the lack of a unit on the vertical scale in fig. 3, the names used in fig. 4a are not in English and should be replaced with their English names.
It is not clear how the values ​​in tables 1-8 are obtained for determining the effects from various binders on soil stabilization, measured by P-wave velocities for the different days of curing time. I recommend that this part of the work be explained in more detail so that it is clear to a wide range of readers, not only to narrow specialists in the field. At the same time, the conclusions are extensive and my recommendation is to optimize them.

Submission Date

18 October 2022

Date of this review

28 Oct 2022 00:14:23

Author Response

Dear Editors,

Please find attached the revised version of the paper. We have carefully followed the comments and critical suggestions and corrected the manuscript accordingly.

All the changes in the text are colored yellow, for the convenience of track changes.

We have proofread the paper throughout and corrected all occasional typographical errors, incorrect punctuation, spelling and minor grammar mistakes, typesetting misprints and inaccurate words where necessary. The replies to the comments in the reviewer’s reports are listed below. Using the opportunity, we thank the reviewer for careful reading of the paper which improved its first version.

With kind regards, - Authors (Per Lind and Polina Lemenkova).

30.10.2022.

Reviewer 2

No

Reviewer’s Comments

Author’s actions

1

I have reviewed the manuscript titled “Dynamics of strength gain in sandy soil stabilized with mixed binders evaluated by elastic P-waves during compressive loading”. Author(s) have investigated the stabilization of binder blended sandy soil through seismic method (compressive loading). They proposed the optimization of binder blending using mixture simplex lattice design with three binders in each case as independent variables and evaluated the effects from single and mixed binders on the gain of strength over time of curing. The research topic is important for development of geo-technology and may be interesting for the research community. The manuscript deserves to be published. However, there are some comments that should be responded before accepting the paper for publication:

General evaluation (no action is required).

2

Authors should mention specifically, in introduction section last paragraph, the novelty of their research.

Added following notation: “Our main contribution can be summarized as follows: 1) the approach of P-wave tests that performs non-destructive measurements of soil strength in selected samples during the period of curing; 2) stabilization using both single and blended binders; 3) using statistical simplex design approach as an associated combinatorial structure and factor experiment for optimization of binder quantity and water ratio; and 4) the use of statistical methods for data analysis based on measurements of P-wave velocities as a function of curing time for individual and blended binary binders.”

Also added the phrase in the previous paragraph: “Therefore, the proposed framework uses seismic measurements which alleviates the 105 above difficulties and disadvantages of of the UCS by measuring soil strength using elastic 106 P-waves. The seismic testing presents an alternative method

3

Authors have followed the existing methodology [ref 60, 61, 62] for soil-binder mixer design and optimization. Authors are advised to add some elaboration of the methodology.

The Methodology is updated, the section itself is reorganized and partially rewritten (news sentences are highlighted in the text). More sentences are added regarding the optimization in the subsection 2.2. Workflow and further in the text.

4

Authors should also include the various basic properties of subgrade soil and binder used in their work in section 3 of MS.

Basic properties of subgrade soil and binder are described and added explanation according to USCS classification, as follows (see sub-subsection 2.1.1. Specimens): "The soil used in this study has a coarse structure and includes fine to medium silty sand and gravel inclusions obtained in the study area of southern Sweden. According to the Unified Soil Classification System (USCS) the sandy soil samples can be classified as follows: clayey sands, sand-clay mixtures (SC) up to 10% of samples, silty sands, sand-silt mixtures (SM) and poorly graded sands (SP) (from 10% to 18% of samples), and well-graded sands, gravelly sands (SW) (18-25% of sample) and belongs to group sands with half or more of coarse fraction, Figure 1. The first group of gravel specimens (25-50%) is presented by clayey gravels and gravel-sand-clays (GC) and silty gravels, gravel-sand-silt mixtures (GM), followed by medium gravel (over 50%), characterized by poorly-graded gravels, 156 gravel-sand mixtures (GP), see Figure 1".

Binders are described in the sub-subsection 2.1.2 Binders: "...we used five different binders for soil stabilization: two traditional binders (cement and lime) and three alternative binders (slag, bio fly ash and energy fly ash). The slag has a type Merit 5000. The first fly ash origin is from Svenska Cellulosa Aktiebolaget (SCA) Lilla Edet and the second is from coal combustion (ISO certified)." The additional paragraph is also inserted in lines 184-193 with more details on GGBFS slag Merit 5000 and bio fly ash.

5

In Figure 5, some terms needs translation in English language.

Corrected: Figure 5 is replotted with text translated into English.

6

Authors should explain the out-put data presented in each column of table 1-8 and should also discuss the negative value obtained for P-wave coefficients.

The negative values obtained for P-wave coefficients correspond to the negative reflection coefficient which implies phase inversion. The seismic reflectivity is raised in terms of P-wave modulus, corresponding to the density of the medium, as reflected in complex seismic traces. This is caused by the viscoelasticity of the soil medium, which presents the effects from the homogeneous background: elastic and viscoelastic perturbation medium. Thus, negative values correspond to the porosity of the selected soil samples.

7

Does the introduction provide sufficient background and include all relevant references? – Can be improved

The Introduction is updated and partially rewritten. Some new phrases and sentences are modified. Added phrases regarding the novelty in the last paragraph with listed contribution and some explanations.

8

Is the research design appropriate? – Can be improved

The research design section is updated and partially modified through reorganization of the subsections. It is now a part of the Methodology section. The subsection 2.2. Workflow is updated (see lines 203-221).

9

Are the methods adequately described? – Can be improved

Methodology is updated and partially modified. Some additional rewordings of the phrases and sentences is made (highlighted in the text). Added live-references to the equations in subsection Optimization of binder mixture.

10

Are the results clearly presented? – Can be improved

The Results section is improved, partially revised and updated with all changes highlighted in yellow colour. Many sentences are rephrased, some more explanations are added where required.

Original Reviewer’s comments

Open Review

English language and style

( ) Extensive editing of English language and style required
( ) Moderate English changes required
( ) English language and style are fine/minor spell check required
(x) I don't feel qualified to judge about the English language and style

Yes

Can be improved

Must be improved

Not applicable

Does the introduction provide sufficient background and include all relevant references?

( )

(x)

( )

( )

Are all the cited references relevant to the research?

(x)

( )

( )

( )

Is the research design appropriate?

( )

(x)

( )

( )

Are the methods adequately described?

( )

(x)

( )

( )

Are the results clearly presented?

( )

(x)

( )

( )

Are the conclusions supported by the results?

(x)

( )

( )

( )

Comments and Suggestions for Authors

Submission Date

18 October 2022

Date of this review

25 Oct 2022 12:34:02

I have reviewed the manuscript titled “Dynamics of strength gain in sandy soil stabilized with mixed
binders evaluated by elastic P-waves during compressive loading”. Author(s) have investigated the stabilization of binder blended sandy soil through seismic method (compressive loading). They proposed the optimization of binder blending using mixture simplex lattice design with three binders in each case as independent variables and evaluated the effects from single and mixed binders on the gain of strength over time of curing.

The research topic is important for development of geo-technology and may be interesting for the research community. The manuscript deserves to be published. However, there are some comments that should be responded before accepting the paper for publication:

  • Authors should mention specifically, in introduction section last paragraph, the novelty of their research.

  • Authors have followed the existing methodology [ref 60, 61, 62] for soil-binder mixer design and optimization. Authors are advised to add some elaboration of the methodology.

  • Authors should also include the various basic properties of subgrade soil and binder used in their work in section 3 of MS.

  • In Figure 5, some terms needs translation in English language.

  • Authors should explain the out-put data presented in each column of table 1-8 and should

also discuss the negative value obtained for P-wave coefficients.

Reviewer 3 Report

In this paper, dynamics of strength gain in sandy soil stabilized with mixed blinders are evaluated using elastic P-waves. To improve the quality of the manuscript, please consider the following comments:

1- Please define all abbreviations in the first position of the paper. There are some undefined abbreviations.

2- What is the difference between experimental workflow and materials and methods? It seems to me that it is possible to present these parts of the manuscript as: 1- methodology: 1-1-workflow, 1-2- concepts, 2- materials, 

3- part 2.2: did you use an optimization method to find optimum parameters?

4- Why elastic P-waves is used? 

5- It seems to me that some materials regarding results section is presented in methodology section. They must be separated. 

6- Figure 5: Please define x-axes. Moreover do not use abbreviations in y-axes.

7- Table: is this number of decimal places significant?

8- Names of columns are not clear! t(47)? Std.Err?

9- Please let me know number of samples used for estimations measures.

Author Response

Dear Editors,

Please find attached the revised version of the paper. We have carefully followed the comments and critical suggestions and corrected the manuscript accordingly.

All the changes in the text are colored yellow, for the convenience of track changes.

We have proofread the paper throughout and corrected all occasional typographical errors, incorrect punctuation, spelling and minor grammar mistakes, typesetting misprints and inaccurate words where necessary. The replies to the comments in the reviewer’s reports are listed below. Using the opportunity, we thank the reviewer for careful reading of the paper which improved its first version.

With kind regards, - Authors (Per Lind and Polina Lemenkova).

30.10.2022.

Reviewer 3

No

Reviewer’s Comments

Author’s actions

1

Please define all abbreviations in the first position of the paper. There are some undefined abbreviations.

Corrected, added abbreviations: Ground Granulated Blast Furnace Slag (GGBFS); Uniaxial/Unconfined Compressive Strength (UCS); Swedish Geotechnical Institute (SGI); Swedish Institute for Standards (SIS); Integrated Circuit Piezoelectric (ICP); PicoCoulomB (PCB) – a trademark; International Organization for Standardization (ISO). They are also added in the Abbreviations section.

2

What is the difference between experimental workflow and materials and methods? It seems to me that it is possible to present these parts of the manuscript as: 1- methodology: 1-1-workflow, 1-2- concepts, 2- materials,

Ok, we reorganized Section 2. Materials and Methods in the manuscript with hierarchical level of the subsections as follows:

2. Materials and Methods

  1. Materials

  • 2.1.1. Specimens

  • 2.1.2. Binders

  1. Workflow

  • Uniaxial Compressive Strength (UCS)

  • P-wave measurements

  • Determination of water content

  • Freeze-thaw tests

  1. Concepts

    • Optimization of binder mixture

    • Process control optimization

    • Total binder optimization

3

Part 2.2: did you use an optimization method to find optimum parameters?

Yes, described additionally in subsection 2.2. Workflow [lines 203-221]. Also, more details are given in section 2.3. Concepts: 2.3.1. Optimization of binder mixture, 2.3.2. Process control optimization, 2.3.3. Total binder optimization.

4

Why elastic P-waves is used?

Because it is a non-destructive techniques which does not crash the sample during testing. We explained it in mode details in the Introduction section with added phrases as follows: “the proposed framework uses seismic measurements which alleviates the above difficulties and disadvantages of of the UCS by measuring soil using elastic P-waves through non-destructive measurements. The seismic testing an alternative method of evaluation of stiffness and strength <...> using measured velocities of elastic pressure waves (P-waves). The core principle of the evaluation of velocities of P-waves consists in the physical theory of waves and elastic properties of soils as a porous media. Thus, measuring P-wave speed enables to estimate the level of stiffness and strength in a soil sample <...>. The technical approach <...> consists in the propagation of elastic waves that penetrate the specimens of soil materials tested at different curing periods. A quantitative assessment of the P-wave velocity gives the value of the soil strength and stiffness due to the correlation between these parameters, which is reported in relevant case studies”.

5

It seems to me that some materials regarding results section is presented in methodology section. They must be separated.

We made the restructuring of the text and moved some sentences and phrases to the Results (always highlighted for track changes). Also, the Section 2. Materials and Methods is partially reorganized.

6

Figure 5: Please define x-axes. Moreover do not use abbreviations in y-axes.

Corrected: X-axis is annotated (Proportions of water, %). Y-axis is explained in full: Number of observations. Also, small note is added on the legend: Effects of curing during 90 days.

7

Table: is this number of decimal places significant?

In the Tables, we used the results obtained from the calculations in the experiments, so we keep the number of decimal places as original values which are significant.

8

Names of columns are not clear! t(47)? Std. Err?

Corrected: t(47) → T=47°C; Std. Err → standard error (abbreviation is explained just below the Table, due to the tight space). Checked and corrected for all the tables. Also, we added footnotes for each table.

9

Please let me know number of samples used for estimations measures.

82 for each case, also included as technical notes below the tables.

10

Does the introduction provide sufficient background and include all relevant references? – Can be improved

The Introduction is updated and partially rewritten. Some new phrases are added, some small rewordings are made in the selected sentences. Added remarks on novelty and contribution.

11

Are the methods adequately described? – Can be improved

The Methodology section is updated with some sentences added, modified or rewritten. Rewordings are always coloured in the text.

12

Are the results clearly presented? – Can be improved

The results section is revised and updated with all changes colored yellow. Many sentences are rephrased, some more explanations are added where required.

13

Are the conclusions supported by the results? – Can be improved

The Conclusion section is shortened: in previous version, it had 1082 words, now – 818.

Original review:

Open Review

English language and style

( ) Extensive editing of English language and style required
( ) Moderate English changes required
( ) English language and style are fine/minor spell check required
(x) I don't feel qualified to judge about the English language and style

Yes

Can be improved

Must be improved

Not applicable

Does the introduction provide sufficient background and include all relevant references?

( )

(x)

( )

( )

Are all the cited references relevant to the research?

(x)

( )

( )

( )

Is the research design appropriate?

(x)

( )

( )

( )

Are the methods adequately described?

( )

(x)

( )

( )

Are the results clearly presented?

( )

(x)

( )

( )

Are the conclusions supported by the results?

( )

(x)

( )

( )

Comments and Suggestions for Authors

In this paper, dynamics of strength gain in sandy soil stabilized with mixed blinders are evaluated using elastic P-waves. To improve the quality of the manuscript, please consider the following comments:

1- Please define all abbreviations in the first position of the paper. There are some undefined abbreviations.

2- What is the difference between experimental workflow and materials and methods? It seems to me that it is possible to present these parts of the manuscript as: 1- methodology: 1-1-workflow, 1-2- concepts, 2- materials, 

3- part 2.2: did you use an optimization method to find optimum parameters?

4- Why elastic P-waves is used? 

5- It seems to me that some materials regarding results section is presented in methodology section. They must be separated. 

6- Figure 5: Please define x-axes. Moreover do not use abbreviations in y-axes.

7- Table: is this number of decimal places significant?

8- Names of columns are not clear! t(47)? Std.Err?

9- Please let me know number of samples used for estimations measures.

Submission Date

18 October 2022

Date of this review

25 Oct 2022 09:39:02

Round 2

Reviewer 3 Report

Accept as it is.